# Examining delays in diagnosis for slipped capital femoral epiphysis from a health disparities perspective

**Maureen Purcell[1], Rustin Reeves[1]\*, Matthew Mayfield[2]**

**1** Department of Physiology and Anatomy, University of North Texas Health Science Center, Fort Worth, Texas, United States of America, **2** Department of Orthopedics, Cook Children's Medical Center, Fort Worth, Texas, United States of America

\* Rustin.Reeves@unthsc.edu

**Data Availability Statement:** We understand that a submission requirement for this journal is to provide a copy of the dataset [for whatever purposes]. However, in order for this study to be

## Abstract

Slipped Capital Femoral Epiphysis (SCFE) is a skeletal pathology affecting adolescents which requires timely surgery to prevent progression. Delays in diagnosis and treatment of SCFE can negatively affect patient prognosis, and few studies have examined how health disparities and barriers to care may influence these delays. In particular, only a handful of studies have included a Hispanic patient sample, despite this ethnic group's increased risk for the disease and unique barriers to care. A retrospective chart review was conducted for 124 patients surgically treated for idiopathic SCFE from January 2010 to September 2017. Patient data included age, facility and date of diagnosis, sex, BMI, race and ethnicity, Southwick slip angle, and insurance type. Results indicated that patients with private insurance were more likely to present with a mild slip than patients who were insured by Medicaid or uninsured, while patients without insurance were more likely to have severe slips. Patients without insurance also had a significantly higher mean slip than patients with insurance. The relationship between insurance status and slip angle degree was significant independent of race, even though Hispanic individuals were significantly more likely to have Medicaid or be uninsured. All patients without insurance, and a majority of those with Medicaid, were diagnosed in the emergency department. Time to diagnosis and slip angle were positively correlated, which suggests that longer delays led to increase of the slip angle, consistent with previous findings. Time to diagnosis and BMI were also correlated, which may be tied to socioeconomic factors, but the possibility of weight bias should not be dismissed. These results suggest that socioeconomic status and other factors may have contributed to barriers to care which led to delays in diagnosis and thus more severe slips. Future SCFE research should include health disparities variables to better inform treatment and prognosis.

## Introduction

Slipped Capital Femoral Epiphysis (SCFE) is a musculoskeletal condition seen in the pediatric population where the physis (or "growth plate") of the proximal femur between the head and

possible, the authors had to enter into a data user agreement with the Institutional Review Board (IRB) at Cooks Children's Healthcare System where data were collected, wherein the site retains all rights and access to these data. Therefore, although the authors are not allowed to include the data, we are happy to provide the appropriate staff with the contact information for the IRB noted above, so that the journal may communicate with the IRB and obtain a copy of these data. The phone number for the Institutional Review Board in the Grants Office is 682-885-2103. Email: CookChildrenResearch@cookchildrens.org.

**Funding:** The author(s) received no specific funding for this work.

**Competing interests:** The authors have declared that no competing interest exist.

neck becomes unstable, and the femoral head slips posteriorly and inferiorly away from the neck of the femur. Because SCFE progresses until the physis fuses, early diagnosis and treatment is key [1–14]. Unfortunately, delayed diagnosis with SCFE patients can occur, often in patients with uncharacteristic pain presentation and public insurance. Far fewer studies exist on the latter risk factor, despite several studies indicating that delaying care due to insurance status is associated with a more severe presentation of diseases in general [15–17], The correlation of insurance status with delays in diagnosis is tied to broader issues of health disparities, but this topic has not been well-examined within the SCFE patient population[15, 16].

Extant research indicates that patients with public health plans, such as Medicaid, experience more barriers to orthopedic care than those with private insurance, especially regarding delays in treatment [18–25]. For example, in many states, patients with Medicaid are required to obtain a referral from a primary care provider before booking an appointment with an orthopedist [19, 22, 26, 27]. Furthermore, doctor's offices are more likely to accept patients with private insurance, and get an appointment sooner, than those with Medicaid. In addition, patients with Medicaid may delay or forgo care due to financial constraints or other barriers [15, 28].

Health disparities among racial and ethnic groups are also observed in orthopedic and surgical practices, and often covary with socioeconomic status [25, 29–31]. SCFE occurs at a relatively higher frequency among certain populations, in particular Black and Hispanic children, who are also more likely to receive public health assistance (e.g., Medicaid or Children's Health Insurance Program [CHIP]) than their White and Asian counterparts [32–34]. The etiology of the varied incidence rates of SCFE in specific populations is contested within the literature, although larger body size and skeletal morphological variations are common theories [33, 35–38]. Despite Hispanic children being at greater risk to develop SCFE, only a handful of studies include a sizable sample of Hispanic patients as a distinct ethnic group, but these do not examine how health disparities may contribute to diagnostic delays [12, 33, 39–43].

The degree of slip severity has been significantly correlated with delays in diagnosis in several studies [1, 2, 5, 7, 9, 13, 44]. A higher slip severity increases the likelihood of premature osteoarthritis and additional surgeries (e.g., osteotomies, and in some cases total hip replacements), whereas a patient diagnosed with a mild slip typically has a better outcome [2, 45–48]. Furthermore, a study from Fedorak et al. (2018) showed a direct association between a longer time from symptom onset to diagnosis and a greater chance of more invasive surgery [1]. This study primarily evaluated SCFE within the scope of how health disparities may affect how quickly SCFE is diagnosed and treated. It should be noted that, to the authors' knowledge, this is the first time Hispanic patients have been represented in a study analyzing associations between insurance status and delays in diagnosis for SCFE. This project was taken in part from a dissertation submitted to the UNT Health Science Center in partial fulfillment for the degree Doctor of Philosophy.

## Methods

A retrospective chart review was conducted for patients surgically treated for SCFE at Cook Children's Medical Center (CCMC). This study included patients who underwent surgery from January 2010 to September 2017. Patients diagnosed with SCFE displayed a Southwick Slip Angle (SSA) of >1˚, and other characteristics of SCFE such as widening of the physis. Subjects were required to be 10–16 years of age at the time of SCFE diagnosis and have at least one anatomic study (such as an x-ray) of the affected hip before and after surgery to be included in the study. Patients with comorbidities that precluded an idiopathic SCFE diagnosis, including radiation therapy, endocrine disorders, renal disease, were not included in the study. The

Cook Children's Healthcare System Institutional Review Board approved this retrospective study on March 02, 2018, and informed consent was waived per board procedure for expedited retrospective studies.

A total of 133 patients were initially selected that received a SCFE diagnosis within the set time frame, with 4 excluded for being outside the age range, and 5 excluded for comorbidities. This left 124 subjects for analysis. Patient data related to identified SCFE risk factors and health disparity variables were collected, including age at diagnosis, facility of diagnosis, time to diagnosis (in weeks), height and weight for all subsequent visits, sex, age, race and ethnicity, height and weight, x-rays to measure slip severity via SSA, and insurance provider. Because of the retrospective nature of the study, not all data points for every subject were available. If the time to diagnosis was given as an estimated period of time (e.g., 3–4 months), a mean of that period would be recorded.

Insurance provider was categorized as private, Medicaid, or none. The facility of the original SCFE diagnosis was gathered from patient notes, but this information was not consistently recorded and therefore not available on all patients. Any visits to medical center of this study prior to the SCFE diagnosis were recorded and noted to be an established patient, as opposed to a new patient, at the time of diagnosis. The non-insurance group was removed for analysis with these two groups, as they are over-represented in the new patient group and may confound results.

The metric of SSA to determine the SCFE grade/severity for each patient was measured by a pediatric orthopedist. It has been established that this measurement has a low degree of intraobserver and interobserver error [49]. The SSA was analyzed as both a categorical variable and continuous variable, as the former is more consistent with previous literature, but the latter provides more information on the spread of variation within the sample. Slip severity was categorized as mild, moderate, and severe. Mild was defined as <30°, Moderate as 31°-60°, and Severe as >61°. BMI-for-age was calculated as a percentile from the CDC website. Race and ethnicity were self-reported from the patient intake form. Thus, "Hispanic" in this study refers to a group of patients who self-identify with the ethnicity and race of Hispanic White. The variable "age at pain onset" was calculated by subtracting the time to diagnosis from the age of diagnosis.

Patient data were gathered from files stored in both Athena and Meditech computer programs at the medical center. Data were recorded in the REDcap data capture program, and then exported and analyzed with SPSS 25 once all identifying information was removed. All ratio data underwent testing for normality and homogeneity of variance and with Kurtosis and Levene's test. A $log_{10}$ or square transformation was used if the data rejected the null hypotheses for normality. Nonparametric tests were used if transformed data did not meet parametric test assumptions.

For normal ratio data, the parametric tests ANOVA, ANCOVA, Pearson's correlation, and linear regression were used. For non-normal ratio data, a Kruskal-Wallis test was utilized. A Tukey's post-hoc test was used for significant ANOVA results. All count data underwent a chi-square analysis, or a Fisher's exact test in the case where at least one cell size was less than 5. Post-hoc tests for count data were conducted using the adjusted residuals and transforming into p-values with a Holm-Bonferroni correction [50].

## Results

Black, White, and Hispanic patients were represented with roughly equal sample sizes, with one excluded from analysis because race was not specified (Table 1). The sample included patients of all three categories of SSA severity, with the highest number in the moderate

**Table 1. SCFE patient demographics and clinical characteristics summary.**

| Demographic |
| --- |
| *Sex* |
| Male |
| Female |
| *Race/Ethnicity* |
| Black |
| Hispanic |
| White |
| *SSA category* |
| Mild |
| Moderate |
| Severe |
| *Insurance Type* |
| Medicaid |
| None** |
| Private |
| *BMI percentile* |
| Normal weight (5th - 85th percentile) |
| Overweight (86th - 95th percentile) |
| Obese (>95th percentile) |

**Eight of the 9 patients in this category filed for Medicaid at time of diagnosis.

category, with two exclusions because of radiograph quality. The highest frequency of patients were initially diagnosed at the emergency department (41.9%), followed by primary care provider offices (25.8%), and then Orthopedist's offices (17.7%). The mean BMI percentile for all patients was 89.78%.

The SSA grade was significantly different across the categories of established vs new patients, and insurance types (Table 2). Post-hoc testing revealed significantly fewer established patients presented with a mild SCFE than new patients (p = 0.01). Patients with private insurance were more likely to present with a mild SCFE (p = 0.006), while patients with no insurance were more likely to present with a severe SCFE (p = 0.006). All other chi-square and Fisher's exact testing examining insurance types and patient category, BMI percentile, and race were not significant.

**Table 2. Categorical analysis of SCFE severity in patient category and insurance type.**

| | SSA category | | | |
| --- | --- | --- | --- | --- |
| | **mild** | **moderate** | **severe** | *p-value* |
| *Patient category* | | | | |
| Established, n (%) | 13 (38.2)* | 15 (64.1) | 6 (17.6) | 0.046 |
| New, n (%) | 15 (17.2) | 54 (44.1) | 18 (20.7) | |
| *Insurance Type* | | | | |
| Medicaid, n (%) | 8 (15.1) | 33 (62.3) | 12 (22.6) | 0.004 |
| None, n (%) | 0 (0) | 4 (44.4) | 5 (55.6)* | |
| Private, n (%) | 20 (23.3)* | 31 (53.4) | 7 (12.1) | |

*Indicates cells with statistically significant difference in post-hoc testing after a Holm-Bonferroni adjustment.

The variables of time to diagnosis, BMI, and BMI percentile underwent log transformations to fit assumptions for parametric testing. Only BMI percentile was still skewed after data transformation, so nonparametric testing was performed for this variable. ANOVAs were run to examine the mean difference in categories within both insurance and race regarding the time to diagnosis, BMI, and SSA (in degrees as a continuous variable). The only significant finding was between groups for insurance type with SSA ($p = 0.003$) (Fig 1). Patients with no insurance had an average SSA of 59.67˚ (high-range, moderate slip), while patients with Medicaid had an average of 44.7˚ (mid-range, moderate slip), and private insurance holders had an average of 38.83˚ (low-range, moderate slip). A Tukey post-hoc revealed that significant differences exist both between the non-insured and Medicaid groups ($p = 0.044$), and between the non-insured and Private groups ($p = 0.003$), with the non-insured group displaying a significantly higher mean SSA than both the Private and Medicaid groups.

Chi-square analyses revealed significant relationships between insurance type and the facility of diagnosis ($p = 0.017$, Fig 2), as well as between insurance type and race/ethnicity ($p = <0.001$, Fig 3). The emergency department was the only facility where patients with no insurance were diagnosed, but post-hoc testing with a Holm correction did not reveal significant differences between Medicaid and private insurance types. In post-hoc analysis for insurance type and race, significant differences were found with White and Hispanic patients, but not Black patients. Fig 3 shows a significantly greater number of Hispanic patients were on Medicaid ($p = <0.001$), while more White patients had private insurance ($p = <0.001$). Conversely, Hispanic patients were significantly less likely to have private insurance ($p = <0.001$), and White patients less likely to have Medicaid ($p = 0.004$).

Because previous chi-square testing revealed significant differences among insurance types within race/ethnicity groups, an ANCOVA was run to determine if any covariation existed

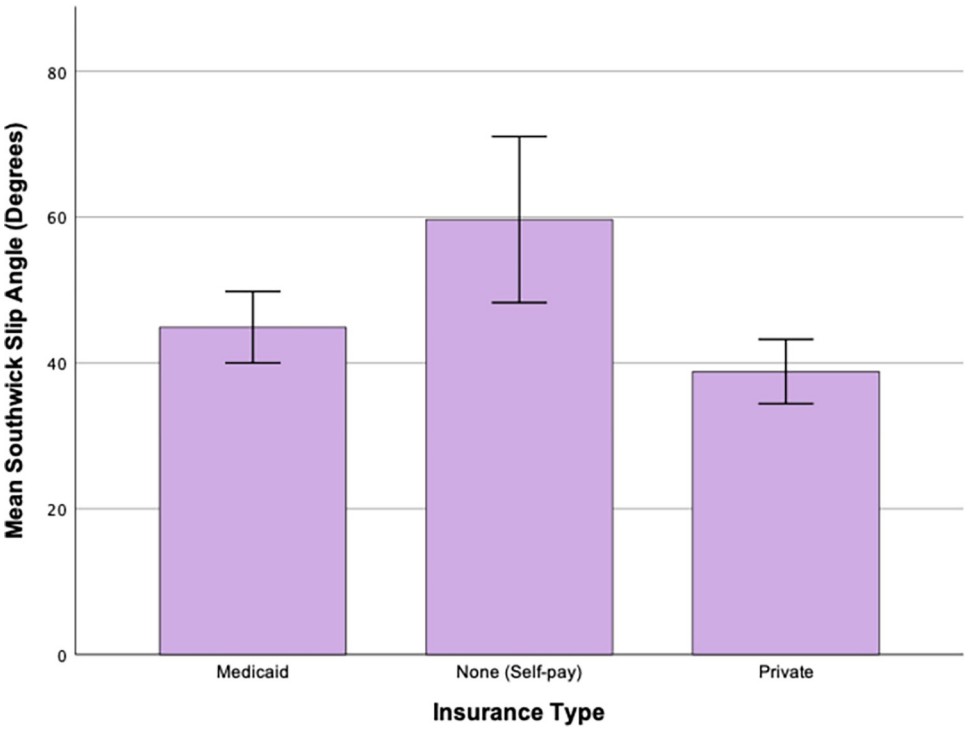

**Fig 1. Southwick slip angle mean separated by insurance type.** Error bars represent 95% confidence interval.

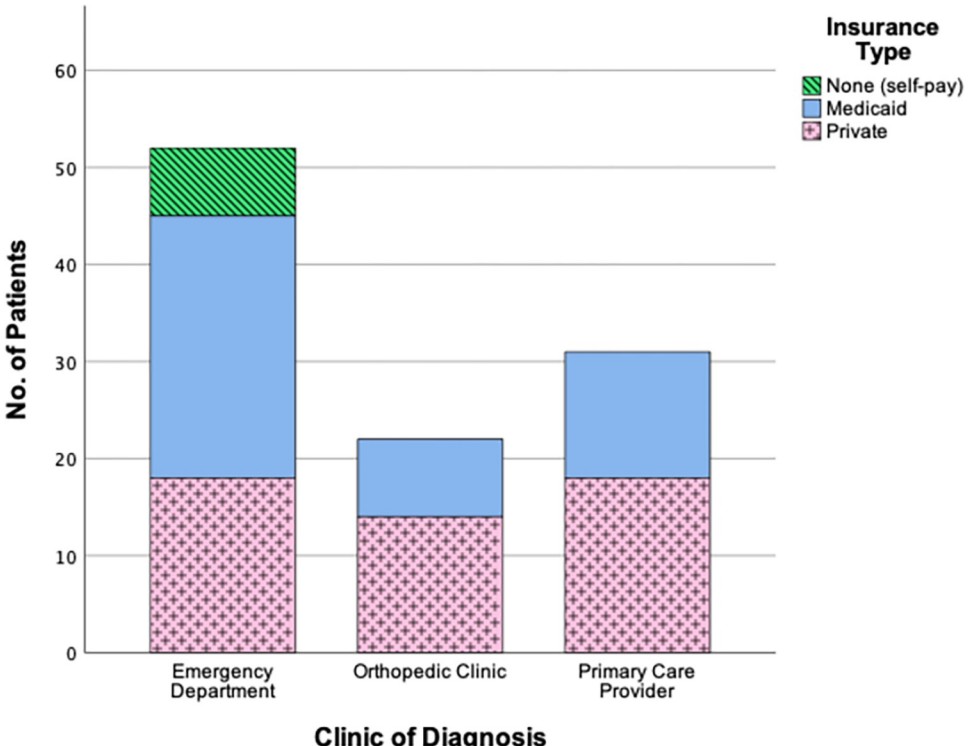

**Fig 2. Clinic of diagnosis for patients with each insurance type.**

with ethnicity and insurance regarding SSA, BMI, and time to diagnosis. None of the results were significant. Nonparametric testing with BMI percentile and time to diagnosis, insurance type, SSA, and SSA categories yielded no significant results.

A Pearson's correlation matrix revealed significant correlations among BMI, time to diagnosis, and SSA. Multiple linear regression analysis also showed significant associations between time to diagnosis and BMI ($r = 0.347$; $p = <0.001$), and time to diagnosis and SSA ($r = 0.321$; $p = <0.001$).

## Discussion

The current study examines what health disparities may exist within the SCFE patient population, and what variables are associated with delays in care and more severe presentations of SCFE. This line of research is also important in evaluating if long-term consequences may exist for patients diagnosed with SCFE who experience barriers to care. To the authors' knowledge, this is the first study to analyze the direct relationship between insurance type and SSA, and only the second to examine delays in diagnosis with an uninsured sample as a distinct group. All patients diagnosed with SCFE require surgery, and how quickly they are able to undergo surgery after symptom onset exerts a direct effect on SCFE severity and hip health outcome [1, 2, 9, 13, 45, 46, 51, 52]. The number of Hispanic patients in this sample was also notable and important for inclusion, considering they are at higher risk for SCFE than White patients [33, 34, 39], and few previous studies have included this ethnic group.

In our study, the non-insurance group had a significantly higher mean SSA than the Medicaid and private groups. Additionally, the categorical analysis of the SSA revealed that patients with private insurance were more likely to present with a mild grade SCFE, and less likely to

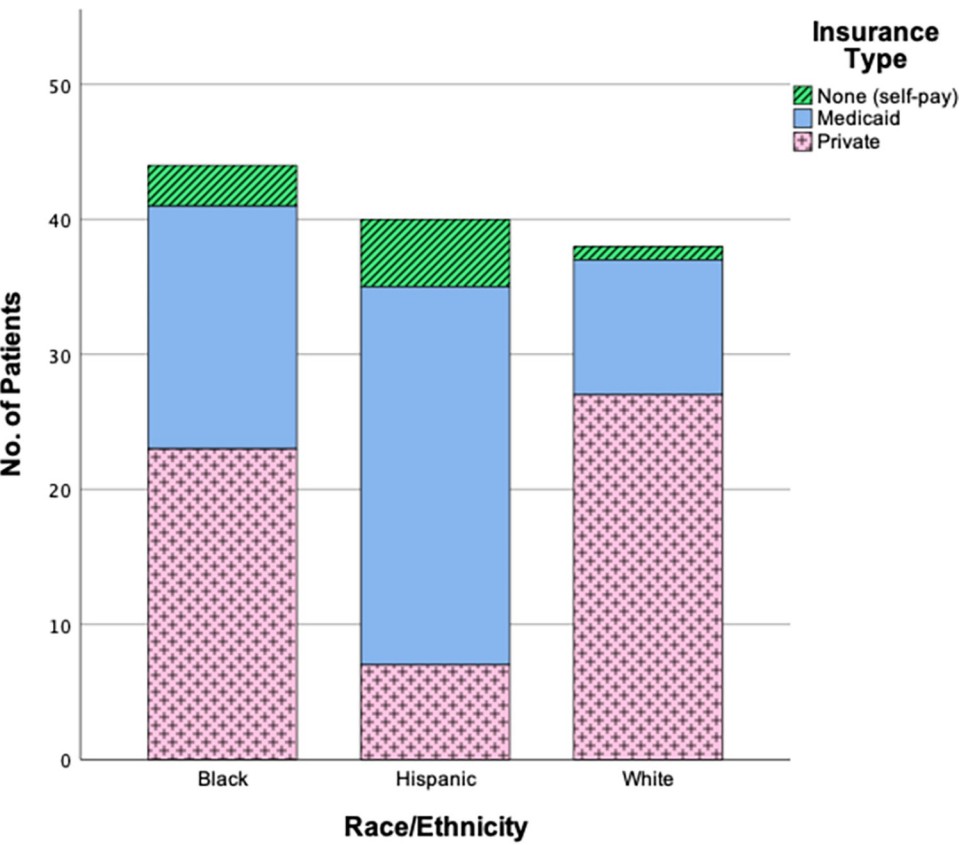

**Fig 3. Insurance types separated by race and ethnicity.**

have a severe SCFE that patients on Medicaid or patients who were not insured. Since delays in diagnosis often progress to a more severe slip, these results suggest that the non-insurance group likely had barriers to care that affected the patients' condition and prognosis, and that patients with private insurance faced fewer barriers to receiving the same treatment. Previous research has noted that access to orthopedic care is often restricted, especially for those on Medicaid [18, 19, 21]. Additionally, orthopedists currently in practice acknowledge that insurance type and socioeconomic status can be a barrier to care [53]. Texas and many other states require patients on Medicaid to get a referral to see an orthopedist–a barrier that patients with Private insurance may not experience. Medicaid recipients also regularly experience delays in acquiring appointments even after receiving a referral. For example, orthopedic clinics are more likely to take private insurance than Medicaid, and patients on Medicaid may be scheduled for an appointment several weeks later than patients with Private insurance [18, 24, 26, 27].

Although a significant difference in the time to diagnosis was not observed among insurance status groups, the time to diagnosis was not available for all patients. Thus, this may have affected the ability to discern a significant relationship among the three insurance status groups. It should be noted that time to diagnosis was a moderate predictor of SSA and showed a positive linear relationship. This is consistent with previous studies [2, 5, 7, 9, 13, 44], and our results suggest, therefore, that a patient's slip severity is positively correlated with delays in care.

The correlations among insurance types and SSA in our sample demonstrate how barriers to care may result in a poorer prognosis for SCFE patients without insurance. Currently, Texas

has the highest rate of uninsured children in the country (partly due to lack of Medicaid expansion) [32]. Severe SCFE is often tied to multiple complications both in the short-term with a greater chance of a more invasive surgery, and over the long-term with a higher risk of osteoarthritis, avascular necrosis, and poorer hip health scores [45–47, 52, 54, 55]. Therefore, examining data of uninsured SCFE patients is particularly important for this area of the country, and our study provides a unique and needed contribution, as well as directions for future research.

Although previous research has noted unique barriers to care that exist within the Hispanic population (e.g. difficulties with acculturation, immigration status, and language) [56–58], no significant differences were found with slip severity among the race and ethnicity categories in our study [56]. Additionally, the ANCOVA analysis revealed that race and ethnicity did not contribute to the differences observed in SSA among insurance types, even though a significantly greater proportion of Hispanic patients were on Medicaid or had no insurance. Our findings suggest that, even though Hispanic patients typically experience unique barriers to care, these factors may dissipate when patients are not underinsured. This aligns with previous studies which attribute a majority of the health disparities observed in Black and Hispanic communities to income and insurance inequality[58, 59]. Alternatively, a change in socioeconomic status (from Medicaid to Private insurance) may indicate an overall lessening of these challenges.

The facility of diagnosis can affect the ability to receive proper care, as primary care providers may have trouble identifying SCFE symptoms, especially when presented as knee pain, and therefore may not believe an immediate referral necessary [2, 7, 9, 12]. Barriers to care may also affect the facility of diagnosis. Previous research has noted that patients insured by Medicaid are more likely to experience transportation difficulties, work schedule conflicts, and barriers obtaining appointments with a primary care or specialist clinic, and thus may choose to go to the emergency department [16, 60–63]. Although post-hoc testing did not show a statistically significant difference between the Medicaid and private insurance groups for the clinic of diagnosis in our data, it should be noted that a majority of patients with Medicaid (51.9%) presented at the emergency department for diagnosis. Additionally, the difference in sample size between the uninsured and insured groups may have skewed the post-hoc testing.

Interestingly, established patients at the medical center were more likely to present with a mild SCFE than new patients. Our data did not indicate a significant relationship between insurance status or race and the new/established patient groups; therefore, it is possible that established patients insured by Medicaid experienced fewer barriers to care than patients insured by Medicaid who were not established patients. Cook Children's Health Care System (HCS) includes several neighborhood clinics intended to serve as medical homes for patients to receive continuous and preventative care, where the staff speak both English and Spanish. Cook Children's HCS also does routine healthcare outreach to help improve healthcare literacy to lower income groups. The medical home model is patient-centered and focuses on comprehensive, continuous care and accessibility by building relationships between patients and providers. Clinics that follow this model are associated with better healthcare outcomes and adherence with patients reporting fewer barriers to care [64–66]. The results of this study suggest that outreach efforts such as those utilized by the Cook Children's HCS reduce barriers to care for the Medicaid population. This correlate should be examined with a more robust sample size and more data before any causal conclusions can be made, especially since a thorough review of the literature did not reveal any studies on delays in SCFE diagnosis that include this variable as a possible correlate to barriers to care.

The linear relationship between BMI and time to diagnosis is noteworthy, and has been reported in only one other study[44]. This correlation suggests that delays in care for SCFE

patients may be more complex than previous research has indicated. Current healthcare providers acknowledge that unfavorable views toward obese patients exist, and multiple meta-review studies have indicated these negative perceptions have persisted for many decades in the healthcare field [67–71]. Obese patients have reported feeling dismissed by their healthcare provider regarding pain concerns [67, 71, 72]. Patients who are obese and encounter weight stigma from their healthcare providers report decreased trust in the health care system, and are more likely to delay or forgo care[73–77]. A higher BMI can correlate with lower socioeconomic status, and result in concerns of treatment cost; however, income information was not available for this sample. The relationship between delays in diagnosis and BMI within the SCFE patient population should be further studied, as this patient population tends to be overweight or obese, and therefore any delays in care regarding this risk factor would be beneficial to address.

Several limitations for this study exist. First, it is a retrospective study, and therefore not every variable was available for each patient because much of the data were self-reported or incomplete (e.g., time to diagnosis and BMI). Second, this is a relatively small study that includes one healthcare system in one state, and thusly may not generalize to other areas of the United States. Third, more nuanced information regarding the private insurance group, such as co-pay, deductible, and organization type, was not available. Therefore, any barriers to care within the private insurance group, such as high copay and deductible and their effect on SCFE diagnosis, were not observable.

The significant relationship between insurance type and SSA (as both a categorical and continuous variable) shows how being underinsured, and in particular uninsured, may negatively affect a patient with SCFE. The health disparities within the SCFE patient population highlight how barriers to care may impact multiple variables and can compound to affect patient treatment and prognosis. Future research could include interviewing patients about their experiences in obtaining an orthopedic appointment to specify and expand on the barriers that exist for this patient population.

## Acknowledgments

We thank Erica Stockbridge, PhD, of University of North Texas Health Science Center in Fort Worth, TX, for her knowledge and insight regarding insurance and barriers to care in marginalized communities.

## Author Contributions

**Conceptualization:** Maureen Purcell, Matthew Mayfield.

**Data curation:** Maureen Purcell.

**Formal analysis:** Maureen Purcell.

**Investigation:** Maureen Purcell, Matthew Mayfield.

**Methodology:** Maureen Purcell.

**Project administration:** Matthew Mayfield.

**Resources:** Matthew Mayfield.

**Supervision:** Rustin Reeves, Matthew Mayfield.

**Validation:** Maureen Purcell.

**Visualization:** Maureen Purcell.

**Writing – original draft:** Maureen Purcell.

**Writing – review & editing:** Rustin Reeves, Matthew Mayfield.

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
