## [Decision Letter · Decision Letter 0]

10 Feb 2022

PONE-D-21-38051Examining delays in diagnosis for slipped capital femoral epiphysis from a health disparities perspectivePLOS ONE

Dear Dr. Reeves

Thank you for submitting your manuscript to PLOS ONE. After careful consideration, we feel that it has merit but does not fully meet PLOS ONE’s publication criteria as it currently stands. Therefore, we invite you to submit a revised version of the manuscript that addresses the points raised during the review process.

We look forward to receiving your revised manuscript.

Kind regards,

Santosh Kumar, PHD

Academic Editor

PLOS ONE

Journal Requirements:

We will update your Data Availability statement on your behalf to reflect the information you provide."

Additional Editor Comments (if provided):

The manuscript was reviewed by two reviewers in the filed. Although they agreed that the manuscript is an important area of research and the study has been done thoughtfully and purposefully, both the reviewers have some important and critical comments. The reviewer 1, in particular, gave several comments that need to be addressed in the revised version. In addition, the discussion needs to include study limitations and future direction in addition to proof-reading for English.

Reviewers' comments:

Reviewer's Responses to Questions

**Comments to the Author**

1. Is the manuscript technically sound, and do the data support the conclusions?

Reviewer #1: Yes

Reviewer #2: Partly

2. Has the statistical analysis been performed appropriately and rigorously? 

Reviewer #1: Yes

Reviewer #2: Yes

3. Have the authors made all data underlying the findings in their manuscript fully available?

Reviewer #1: Yes

Reviewer #2: Yes

4. Is the manuscript presented in an intelligible fashion and written in standard English?

Reviewer #1: Yes

Reviewer #2: Yes

5. Review Comments to the Author

Reviewer #1: Feedback/Suggestions/Comments

1. Change terminology from “physician” to primary care provider or healthcare provider throughout the manuscript to be more inclusive of other healthcare providers such as advanced practice registered nurses (nurse practitioners) and physician assistants.

1. There are minor grammatical errors throughout the document that need to be reviewed and revised.

2. In the introduction section this question should be answered: Why are black and Hispanic children more likely to be diagnosed with SCFE? It will be important to include this information for readers who are not as well versed in SCFE and are reading this manuscript because of their interest in the impact of health disparities on healthcare outcomes. It also reinforces the information listed in lines 197-200 and 243.

3. Can the first sentence of the discussion section be added to the introduction as part of the purpose of the study?

4. I really appreciate how the authors included why the location of this study (Texas) was important to include in the manuscript.

5. Starting with line 218, I wanted the authors to provide examples of the barriers to care that were referred to. However, I realized that the examples were in the next paragraph. For continuity of content/flow of content, I recommend either combining the paragraph that starts with line 222 to the preceding paragraph or consider at sentence at the end of the end of line 221 to prepare readers for the barriers that you’re going to discuss in the upcoming paragraphs.

6. In line 240, can you make the connection to health disparities?

7. Lines 258-259 “Healthcare providers can also carry unfavorable bias toward obese patients” resulting in? (Consider adding more information that highlights the impact of obesity bias on healthcare outcomes)

8. I do not understand the relevance of including line 84 if this is a standard course of action. Including this line may also prompt readers to wonder why there was one exception. I recommend removing line 84.

9. Table 2 is very well constructed/easy to follow and understand

10. I appreciate the information that was included in lines 268-271

11. In line 275, are there any other possible reasons that patients who are uninsured or covered by Medicaid may be diagnosed in emergency departments that can be supported by the literature?

12. Overall, very well written discussion section that highlights key points from the results

13. Succinct abstract, well-written. Highlights key points

14. Although this may drastically increase your word count, I still recommend the use of “people-first language” throughout the manuscript. For example, instead of “Medicaid patients,” consider changing to patients insured by Medicaid. There are also several other opportunities to include people first language throughout the manuscript.

Reviewer #2: In this study, authors examining delays in diagnosis for slipped capital femoral epiphysis from a health disparities perspective. Proofreading the manuscript is a must to avoid errors in writing. Ensure mention of the limitations of the study within discussion, along with future directions. The discussion can be improved and more centered.

6. PLOS authors have the option to publish the peer review history of their article (what does this mean?). If published, this will include your full peer review and any attached files.

Reviewer #1: No

Reviewer #2: **Yes: **Vivek Kumar Kashyap

---

## [Author Response · Author response to Decision Letter 0]

27 Apr 2022

We appreciate the comments and concerns from both reviewers. Thank you for your time and assistance with reviewing the manuscript. In addition to the reviewer’s comments/concerns, we placed the two tables back-to-back in the manuscript, and then slightly changed the order of the figures. Every figure/table was kept the exact same as well as the results for the tables/figures; however, we did feel the order they are in now makes it easier for the reader to follow. Again, we are grateful for the extremely thorough review of our manuscript and thankful for the time and energy put in by the reviewers and editorial team to support this study.

---

## [Decision Letter · Decision Letter 1]

27 May 2022

Examining delays in diagnosis for slipped capital femoral epiphysis from a health disparities perspective

PONE-D-21-38051R1

Dear Dr. Reeves

We’re pleased to inform you that your manuscript has been judged scientifically suitable for publication and will be formally accepted for publication once it meets all outstanding technical requirements.

Kind regards,

Santosh Kumar, PHD

Academic Editor

PLOS ONE

Additional Editor Comments (optional):

Reviewers' comments:

Reviewer's Responses to Questions

**Comments to the Author**

1. If the authors have adequately addressed your comments raised in a previous round of review and you feel that this manuscript is now acceptable for publication, you may indicate that here to bypass the “Comments to the Author” section, enter your conflict of interest statement in the “Confidential to Editor” section, and submit your "Accept" recommendation.

Reviewer #1: All comments have been addressed

Reviewer #2: All comments have been addressed

2. Is the manuscript technically sound, and do the data support the conclusions?

Reviewer #1: Yes

Reviewer #2: Yes

3. Has the statistical analysis been performed appropriately and rigorously? 

Reviewer #1: Yes

Reviewer #2: Yes

4. Have the authors made all data underlying the findings in their manuscript fully available?

Reviewer #1: No

Reviewer #2: Yes

5. Is the manuscript presented in an intelligible fashion and written in standard English?

Reviewer #1: Yes

Reviewer #2: Yes

6. Review Comments to the Author

Reviewer #1: Thank you for changing to people-first language throughout the manuscript. However, there are about 3 times in which "Medicaid Patients" is still listed, starting with the abstract. I'm not sure if this was intentional, but if not, I just recommend these minor changes before the final submission.

Reviewer #2: The author did a detailed response to all reviewer comments. Proofreading the manuscript is a must to avoid errors in writing.

7. PLOS authors have the option to publish the peer review history of their article (what does this mean?). If published, this will include your full peer review and any attached files.

Reviewer #1: No

Reviewer #2: **Yes: **VIVEK KUMAR KASHYAP

---

## [Editor Report · Acceptance letter]

10 Jun 2022

PONE-D-21-38051R1 

Examining delays in diagnosis for slipped capital femoral epiphysis from a health disparities perspective 

Dear Dr. Reeves:

I'm pleased to inform you that your manuscript has been deemed suitable for publication in PLOS ONE. Congratulations! Your manuscript is now with our production department. 

Kind regards, 

on behalf of

Dr. Santosh Kumar 

Academic Editor

PLOS ONE